# Comparison of General Anxiety among Healthcare Professionals before and after COVID-19 Vaccination

**DOI:** 10.3390/vaccines10122076

**Published:** 2022-12-05

**Authors:** Zohair Ali Badami, Hareem Mustafa, Afsheen Maqsood, Soha Aijaz, Sara Altamash, Abhishek Lal, Sara Saeed, Naseer Ahmed, Rahima Yousofi, Artak Heboyan, Mohmed Isaqali Karobari

**Affiliations:** 1Department of Prosthodontics, Altamash Institute of Dental Medicine, Karachi 75500, Pakistan; 2Department of Oral Pathology, Bahria University Dental College, Karachi 74400, Pakistan; 3Department of Orthodontics, Altamash Institute of Dental Medicine, Karachi 75500, Pakistan; 4Research Development and Review Cell, Altamash Institute of Dental Medicine, Karachi 75500, Pakistan; 5Department of Prosthodontics, Faculty of Stomatology, Yerevan State Medical University after Mkhitar Heratsi, Str. Koryun 2, Yerevan 0025, Armenia; 6Conservative Dentistry Unit, School of Dental Sciences, Universiti Sains Malaysia, Health Campus, Kota Bharu 16150, Malaysia; 7Department of Conservative Dentistry & Endodontics, Saveetha Dental College & Hospitals, Saveetha Institute of Medical and Technical Sciences University, Chennai 600077, Tamil Nadu, India

**Keywords:** coronavirus, vaccines, COVID-19 pandemic, dental clinics, dental anxiety

## Abstract

Vaccination plays a crucial role in controlling the rate of coronavirus transmission and infectivity. Healthcare professionals are, in fact, at the greatest risk of contracting coronavirus due to their proximity and prolonged exposure to infected patients; this certitude alone enhances the stress and anxiety among patients and professionals alike. In this study, we aimed to assess the levels of anxiety experienced by healthcare professionals in their practices before and after getting vaccinated. This cross-sectional study was carried out in 2021. An electronic survey was distributed among the non-vaccinated and vaccinated healthcare workers. The survey consisted of the following parts: demographic characteristics, coronavirus-related questions, questions related to the specific field of healthcare professions, general anxiety questions, and working-hour-related questions. The Modified General Anxiety Scale (GAD-7) was used along with the paired t-test, Mann–Whitney U test, and Spearmen’s test for comparison. *p* ≤ 0.05 was considered statistically significant. A total of 798 healthcare professionals participated in the study. In this study, the majority of participants were females, with 598 (74.9%) being between the ages of 21 and 30, and 646 (80.9%) participants were graduates, with the majority being dentists. Non-vaccinated healthcare professionals had severe anxiety (30.9%), whereas, in vaccinated participants, anxiety levels were minimal (56.9%). A statistically significant correlation was discovered when comparing the scores of the vaccinated and non-vaccinated individuals as well as when comparing the professions of vaccinated participants, whereas no association was found with the gender and education level of participants. Vaccination is necessary for all entitled individuals to control the spread of coronavirus. It was discovered that there was an increase in anxiety levels before the vaccination was introduced. The anxiousness was greatly lessened following mass immunizations. Our research will help to raise public awareness of stigmatized mental health disorders in the healthcare industry.

## 1. Introduction

The infamous coronavirus outbreak has indeed wrecked the entire world. The virus was first detected in China in December 2019 and has rapidly marked its territory worldwide. Not long after, it was declared a global health crisis.

Coronavirus is spread through direct transmission and contact via respiratory droplets and mucous membranes of the eyes and nose [1,2,3]. Because of the spontaneous arrival of this virus, immediate action had to be taken to control the spread. Healthcare workers were at the greatest risk of contracting the virus, especially dentists, since procedures related to the oral cavity result in the formation of aerosol droplets, increasing the risk [4,5,6]. This made routine procedures troublesome in addition to causing enormous stress and anxiety among healthcare workers [3,7]. Anxiety is related to a decrease in the quality and quantity of treatment [8,9] that results in the patients’ as well as the practitioner’s well-being being compromised [10,11]. Interestingly, several studies concluded that among all professions, dentistry has the highest rate of stress generated [10,12,13,14], and now, due to COVID-19, the stress levels have been further elevated. To combat this issue, the World Health Organization (WHO) released important guidelines that were introduced that included the use of personal protective equipment (PPE) while treating patients. Moreover, elective procedures were put on hold while emergency procedures were being performed [9,15] after PCR testing. In late December 2020, after thorough research, COVID-19 virus vaccines were introduced with the aim of mass vaccination to control its rapid spread and were approved by the World Health Organization (WHO) [4,6]. Initially, the vaccines were available for all healthcare workers, and then they were gradually introduced to the general population. The importance of vaccines was accepted by many people, but there were also people that were resistant to getting a vaccine due to factors such as certain religious views and a lack of understanding of science and the healthcare profession [6,16]. The entire concept of vaccination was not to eliminate the signs and symptoms but to reduce the severity of the effects of the virus. However, as the population started to get vaccinated, there was a considerable decrease in the cases of the COVID-19 virus; this eased both the patients’ and the healthcare professionals’ anxieties. Elective treatments resumed, and the healthcare workers became more comfortable performing procedures after the campaign for vaccination began. Within a year, things started going back to normal with proper protocols being taken.

A few studies were conducted during the COVID-19 pandemic when immunizations were not available and related research was only beginning. As a result, the accuracy of measured anxiety levels was dependent on assumptions rather than actual findings. Many studies overlooked the responses of those who were already suffering from anxiety, and how the COVID-19 pandemic might have aggravated their worry. By questioning healthcare workers about their perception regarding immunizations, our study attempted to reduce error gaps. Given the widespread transmission of misinformation, it was critical to determine if the participants truly comprehended the importance of vaccination. Our study also included in-depth questions on how the immunized and unimmunized participants felt when interacting with patients. How anxious/nervous/irritable were they? This was included in our questionnaire so that we could obtain a detailed and accurate response on how the participants personally felt. This study also included in-depth questions on how the participants felt when interacting with patients. We aimed to assess the levels of anxiety and stress levels among healthcare professionals, especially dentists, both with and without having received the COVID-19 vaccination. We hypothesized that post-vaccination, they had an optimistic approach regarding treating the patient due to a reduction in anxiety levels.

## 2. Materials and Methods

### 2.1. Study Design and Sample Size

A descriptive, cross-sectional survey-based study was carried out from February to May 2021 in Karachi, Pakistan. The ethical review committee of Altamash Institute of Dental Medicine, Pakistan, granted the ethical approval (AIDM/ERC/02/2021/02). This study was executed following the principles of the Declaration of Helsinki. Healthcare professionals residing in Karachi, Pakistan, were invited to participate in this study using the non-probability convenience sampling method. The purpose and objective of the study were explained to the participants. The data were collected after obtaining informed consent from participants through E-mails; to ensure voluntary participation, written and verbal consent was obtained from all the participants. The anonymity of the participants’ data was maintained throughout this study.

An online well-structured questionnaire was designed using Google© forms and distributed to the participants through social media platforms such as Facebook©, WhatsApp©, and E-mail to groups comprising healthcare professionals such as doctors, nurses, and so on, as well as via referrals of healthcare professionals from previous connections. Furthermore, we selected healthcare professionals who actively offered treatment at various hospitals and clinics for this study. Since the coronavirus pandemic situation at the time did not allow us to interact personally with the participants, soft copies of the questionnaire were preferred. The questionnaire was distributed to healthcare professionals practicing in both private and public sectors of Karachi, Pakistan. The questionnaire was filled in by the identified vaccinated and unvaccinated participants via E-mails. Using the Open-Epi software, the sample size of this study was calculated. Keeping the confidence interval at 95% and desired percentile at 50, the total sample size was calculated to be 798. The number of participants in each vaccinated and non-vaccinated group was 399, respectively, n = [(DEFF ∗ Np(1 − p)]/[(d2/Z21 − α/2 ∗ (N − 1) + p ∗ (1 − p)].

### 2.2. Questionnaire Design and Distribution

The questionnaire consisted of 5 parts, as follows: the first section included demographic characteristics such as age, gender, education, and occupation. The second section included questions related to participants’ specific profession in the healthcare system, such as doctors, dentists, and others and if they were suffering from any significant medical conditions (diabetes, hypertension, comorbidity, etc.). The third section included whether or not the participants had been exposed to coronavirus and what perceptions they had of the vaccination setup during COVID-19, such as whether it was a necessary means of preventing the further spread of the virus or not. Next, the general anxiety of the participants was logged while treating/counseling patients. Participants who were not vaccinated were asked how often they experienced anxiety and/or worry while interacting with patients. Furthermore, inquiries were made about whether or not they became easily agitated while working with patients, and the responses of individuals who had been vaccinated were recorded. The latter set of questions featured the same inquiries as the previous ones. Lastly, inquiries such as the work hours per day were noted and whether or not safety precautions were taken to prevent the contraction of the virus. This section further included the fear of contracting COVID-19 among unvaccinated and vaccinated individuals. The questionnaire was formulated in the English language and translated into Urdu for some individuals. The questionnaire was self-administered to overcome the issue of biasness whilst distributing the questionnaire, and duplicate forms were removed.

### 2.3. Anxiety Analysis

The Modified General Anxiety Scale (GAD-7) consists of seven questions, each of which assesses general anxiety levels in various medical settings. Every question is answered on a 3-point Likert scale with responses ranging from “Not at all” to “Nearly every day”, as shown in Figure 1. Each response is given a score between 0 and 3. As a result, a “not at all” reaction receives a 0, and a “nearly every day” one receives a 3. The sum of response scores from all seven questions is used to determine the healthcare professional’s level of anxiety. This scale has a total score of 0 to 21, with cut-off scores of 15 to 21 showing severe anxiety.

### 2.4. Inclusion and Exclusion Criteria

Inclusion criteria:Healthcare professionals such as nurses (specialized training in providing care for the sick or elderly), dentists, and medical doctors;Healthcare professionals who were practicing during the COVID-19 pandemic before and after getting vaccinated.

Exclusion criteria:General population;Dental assistants (assist the dentist during dental procedures and hold dental instruments) and technicians (construct patient prostheses such as dentures and bridges);Paramedics (experts with specialized training in emergency treatment);Pharmacists (specialized in the preparation, storage, and distribution of medications) and physiotherapists (experts who focus on treating injuries that affect movement).

### 2.5. Statistical Analysis

Analysis was performed using the SPSS software (IBM Corporation, SPSS Inc. Chicago, IL, USA v.24) to calculate the mean, frequency, percentage, and standard deviation of demographic data. Paired t-test was used to compare the Modified General Anxiety Scale (GAD-7) scores before and after vaccination and the Mann–Whitney U test was used to correlate them with gender; to compare demographic traits with anxiety levels before and after vaccination, a Spearman’s correlation test was opted for. *p* ≤ 0.05 was considered statistically significant.

## 3. Results

In this study, out of a total of 798 participants, 399 were vaccinated, whereas the other half were not; the majority of participants were females, with the remainder being males. Amongst the vaccinated HCP, 295 (73.9%) were in the age bracket of 21–30 years. Forty-two participants belonged to the 41–50 years group. However, 27 (6.7%) patients were 31–40 years old and 25 (6.2%) were above 50 years of age. Additionally, only 16 (2%) participants were from the 10–20 years category. The mean age of study participants was 24.32 ± 0.281. Additionally, in the non-vaccinated group, 303 (75.9%) participants were in the 21–30 years bracket and 34 (8.5%) belonged to the 41–50 years category. The mean age of no-vaccinated participants was 26.41 ± 0.493. Regarding the level of education, an overwhelming number of the participants were graduates, with a small proportion pursuing postgraduate studies, and the majority of the participants were dentists, followed by medical doctors and nurses, as shown in Table 1.

Regarding medical history overall, 606 of the participants (76%) did not suffer from any significant medical condition, and those who did had the highest incidence of respiratory diseases such as asthma preceded by diabetes. Regarding being tested for coronavirus, 766 (55%) had tested positive at one point, and all the participants unanimously voted that vaccination is necessary to prevent the spread of coronavirus. When polled, 304 (38%) of healthcare employees reported working more than 5–10 h per day, with 264 (33%) working 1–5 h per day. After being surveyed, 718 (89.8%) healthcare professionals said they took all precautionary measures, such as the use of PPE to keep the environment safe from contagious diseases.

Regarding the anxiety experienced by the health professionals in their workplace, nearly one-fourth of the non-vaccinated healthcare professionals felt on edge every day while dealing with patients (Table 2); nevertheless, for the vaccinated participants, the general scores were minimal for more than half of the HCPs (Table 3). GAD-7 overall anxiety ratings were trending toward the severe category for the unimmunized, but for the immunized participants, the graph plummeted into the minimum and mild categories. It was further noted that for unvaccinated participants, the GAD-7 scores remained fairly high throughout the study, in contrast to those of vaccinated participants, which remained relatively low, as presented in Figure 2. As for being easily irritable whilst dealing with patients due to the fear of coronavirus, 92 (23%) of the unvaccinated participants suggested experiencing such a fear, while amongst the vaccinated, it was reported that 223 (56%) had no feeling of easy irritability.

The paired *t*-test revealed a statistically significant difference in GAD-7 scores between the vaccinated and unvaccinated groups according to Table 4. Regarding gender, using the Mann–Whitney U test, a statistically significant relationship was noted between gender and the GAD-7 score; females were more concerned regarding contracting the coronavirus as compared to males, as shown in Table 5.

The analysis of independent variables (gender, education level) showed no correlation with GAD-7 scores amongst vaccinated and unvaccinated participants, whereas a strong correlation (rho = 0.084) was found between the profession and GAD-7 scores of vaccinated participants, as presented in Table 6.

## 4. Discussion

The pandemic did inflict significant economic damage, but it also began to have an impact on mental health. Anxiety was one of the most evident catastrophic impacts that appeared to be triggered by the coronavirus epidemic. Our research was carried out to assess the levels of anxiety among vaccinated and unvaccinated healthcare personnel. We found that elevated levels of anxiety were seen before vaccinations were introduced; these levels declined with the development of vaccines. The GAD-7 scores for most of the unvaccinated participants were positioned in the severe category since health workers were afraid to work due fear of encountering this dangerous life-taking virus, but soon after vaccinations began, the percentage of scores in the severe category tumbled to 2% as vaccination had mitigated workers’ fear of the virus. Healthcare workers were the most affected by the continuously rising number of reported cases of COVID-19. The virus’s quick emergence necessitated dealing with patients consecutively, which elevated psychological stress [17,18]. Anxiety was seen as a critical health concern since it affected not only the healthcare professional but also the patient’s treatment results and well-being. Most of the unimmunized participants were experiencing severe anxiety compared to the vaccinated participants, who experienced minimal anxiety as soon as the immunizations were administered. Aside from mild anxiety, other behavioral changes showed a significant difference. When asked if they had any problems dealing with patients, getting annoyed easily, or relaxing, more than half said they did not have this issue. Vaccines not only assisted in damage control around the world but also in the improvement of human behavior by allowing healthcare professionals to work freely in the field of their choice, as they had previously. It was also discovered that coronavirus cases had fallen marginally and that many patients were suffering less severe symptoms even after the arrival of coronavirus subvariants. Nevertheless, PPE was still advocated as an added protective measure to further limit dissemination; however, its usage appeared to decrease after the vaccinations were introduced, as mentioned previously. A statistically significant association between GAD-7 scores amongst the vaccinated and unvaccinated participants and demographic data led to the conclusion that these independent characteristics, such as females being more affected by anxiety, might have altered the total GAD-7 score. According to our demographic data, females showed higher degrees of anxiety, indicating that females were more emotionally impacted by the pandemic. The majority of participants were young people, indicating that they are cognitively better able to deal with stressful conditions. Dentists were more worried about getting the virus since they were in close contact with patient aerosols. As the immunizations were implemented, healthcare workers felt more secure than before, and the quality and quantity of their patient care improved dramatically. However, the vaccine did not entirely remove anxiety, and other factors, such as day-to-day struggles, must be considered.

Various studies showed that the number of COVID-19 cases decreased after the arrival of the vaccines, but one of these studies revealed that 3.3% of vaccinated healthcare workers had still managed to contract the infection [17,19]. With that said, the objective of vaccination was to lessen the intensity of the symptoms, not to cure the condition. It was also discovered that the utilization of personal protective equipment reduced following immunization. In a study conducted in Turkey among dental professionals, it was found that in unvaccinated dentists, the average mean number of PPE used was 4.69; following vaccination, the average was 4.3 [17]. The reason behind this decline could be the increase in their trust in the vaccine’s effectiveness. A study that was carried out in Italy reported that dentists were the most severely affected by the pandemic among all healthcare workers. A total of 85% of participants conveyed fear of infection during dental procedures, while 70.2% stated that they had anxiety due to the outbreak [17]. A study by Ahmed M.A et al. concluded that 78% of dentists were suffering from fear and anxiety during the pandemic [20]. According to one study, it was discovered that, along with anxiety, healthcare professionals had developed a sleep disorder during the pandemic [21,22,23,24]. This, in turn, was linked to psychological disturbance [22,25]. Another study [22] found that having high-risk family members was an independent risk factor for anxiety. The cause of the increased levels of anxiety was that the healthcare professionals feared they might inadvertently infect their relatives. An additional study supported this finding [26]. UK-based research on ICU doctors carried out in 2018, before the COVID-19 epidemic, concluded that 16% of the staff experienced major anxiety and 8% faced major depression. An increase in these numbers was seen during the pandemic in this specific group of healthcare workers [27]. Another unexpected finding was that professionals in healthcare who had depression before the vaccines were introduced and also had children had lower scores than childless people. This could have been a result of their compassion and connection to their child, which aided greatly during the pandemic phase [28,29,30,31,32]. A gender difference was also noted; looking at other studies, it appeared that females had higher levels of anxiety than men during the pandemic. This is supported by evidence that females exhibited elevated levels of symptoms in particular [33,34]. In contrast to other research findings, it was discovered that experts in healthcare in Lebanon experienced comparatively less anxiety during the pandemic, with only 23% of Lebanese healthcare professionals showing signs of anxiety. This could be because they had a calmer environment, sufficient training, and proper PPE [35]. Contrary to our expectations, every single participant agreed that vaccination is an effective method of controlling the virus’s spread. Social media have grown massively, and during the pandemic, a lot of false information/myths about the vaccine’s arrival were propagated, for example, that vaccines may be hazardous to humans. Many other studies, such as [33,36], concluded that vaccinated people had lower anxiety levels and were happier with their lives. The COVID-19 pandemic vastly affected the quality of life of healthcare workers, which was directly related to their mental health. Hacimusalar et al. discovered that there was a positive correlation between anxiety and hopelessness among these workers. A rise in anxiety levels could account for the 29% increase in hopelessness levels [36]. Anxiety and hopelessness were statistically greater in individuals whose sleep and eating patterns were disrupted by the pandemic and those who dreaded working in hospitals during that period [37]. The reason behind this correlation could be that healthcare workers were working long hours during the day and not having any idea of what would happen next. This resulted in a domino effect that leads to a lack of sleep, making minor errors in diagnosis, fatigue, and ultimately mental and physical burnout [33]. According to our data, the GAD-7 score was significantly higher, especially in unvaccinated individuals. With this, we can predict that since the anxiety levels of these workers were higher, their level of uncertainty and hopelessness was also higher. Hopelessness is thus a positive predictor of anxiety [33]. With the emergence of the virus, the strict use of personal protective equipment (PPE) was stressed. Our findings on vaccination reducing anxiety levels are coherent with a study that found that the prevalence of anxiety was 15.6% and 31.9% among vaccinated and unvaccinated Bangladeshi healthcare workers, respectively, during the COVID-19 epidemic [38].

This study aims to make the general population aware of the correlation between anxiety and vaccination among our healthcare professionals. This decreases the stigmatization associated with not only mental health but also the fear associated with vaccines; our results prove that there was a reduction in anxiety levels post-vaccine. As far as we know, this is true for both vaccinated and unvaccinated healthcare workers. We were able to obtain optimum scores that were relevant to our study since we employed the GAD-7 scale based on seven questionnaires to measure anxiety levels. We relied on several statistical methods to investigate various correlations throughout the investigation. Our study was carried out following the implementation of immunizations. This provided us with a dependable source of information rather than findings based on their interpretation. As with past research, ours had limitations. To begin with, because this was a cross-sectional study, proving causality was difficult. Secondly, because this study was conducted during the pandemic, we did not have enough data to compare the findings for anxiety before the pandemic and before vaccinations were introduced. We were unable to reach out to people from lower socioeconomic backgrounds because our survey was based on an online questionnaire, although we did try to translate the questionnaire into Urdu for a few respondents. Moreover, because the participants had to complete the questionnaires independently, we could not guarantee reliability. Our sample size was insufficient, resulting in limitations since it was skewed toward younger age groups, with females dominating and focusing mainly on dentists based in Karachi. Broader generalized coverage was not possible. Confirmation bias is also a potential limitation as the information was derived from a survey using online questionnaires, which could have led to the participants interpreting the questions in a way other than that intended. The lack of other confounding variables, such as people with psychiatric disabilities/medical, dental assistants, technicians, and the general population, is also a limitation of our study. An increase in the number of questions usually leads to a reduced attention span amongst participants, especially in an online survey like ours. Our study did not compare healthcare professionals who work in private and public settings. This offers the possibility of adding more survey questions, justifying our findings, and making a fair comparison. There are many unanswered questions concerning certain specialties in these professions as some are associated with higher anxiety levels than others. There could have been questions about the personal lives of these professionals, i.e., marital status/number of children, as these are important contributors to one’s mental health. Other questions we could have asked would be regarding any (diagnosed) psychological problems the participants might have, for example, PTSD, OCD, etc. Additionally, participants could have been asked about the number of patients they examined per day. Future studies should be longitudinal to draw more conclusive results so that anxiety levels could be monitored over a longer period. Since this study was only limited to healthcare workers, we cannot generalize the results to the whole population; further investigation is needed to derive results for the general population. In the future, surveys should be supplemented by interviews to avoid participant exhaustion and a reduced attention span. More research is needed to assess the effects of vaccination on anxiety levels in a private vs. public situation, which would then incorporate additional aspects such as working hours, degree of interest, motivation, and so on. These elements could potentially be linked to anxiety. In future research, more cities in Pakistan and other nations should be included to gain a deeper understanding of the findings. The aim of our research was to make people aware of mental health issues prevalent in the healthcare field, which are generally stigmatized. It also opens doors for potential future research. The proper protocol should be established by the government for healthcare professionals to reduce their stress levels in the future, even if vaccines are not available. A committee of physiatrists should be established to assess personnel’s mental fitness before dealing with patients, particularly if another wave or pandemic occurs in the future.

## 5. Conclusions

With these preliminary data, we can conclude that both vaccinated and unvaccinated healthcare staff experienced anxiety as a result of the pandemic; however, the incidence was lower among the vaccinated. It is evident that immunization has produced a relatively safe environment for healthcare professionals, resulting in reduced stress and worry when dealing with patients. This way, healthcare personnel will feel less nervous when dealing with a patient. The quality of service would improve if they could devote their full attention to the treatment rather than continuously fearing for their lives. This study confirms that measures should be implemented to vaccinate as many healthcare professionals as possible to limit the risk of infection and anxiety. However, because SARS-CoV-2 has not been completely eradicated, all preventive measures must be strictly followed.

## Figures and Tables

**Figure 1 vaccines-10-02076-f001:**
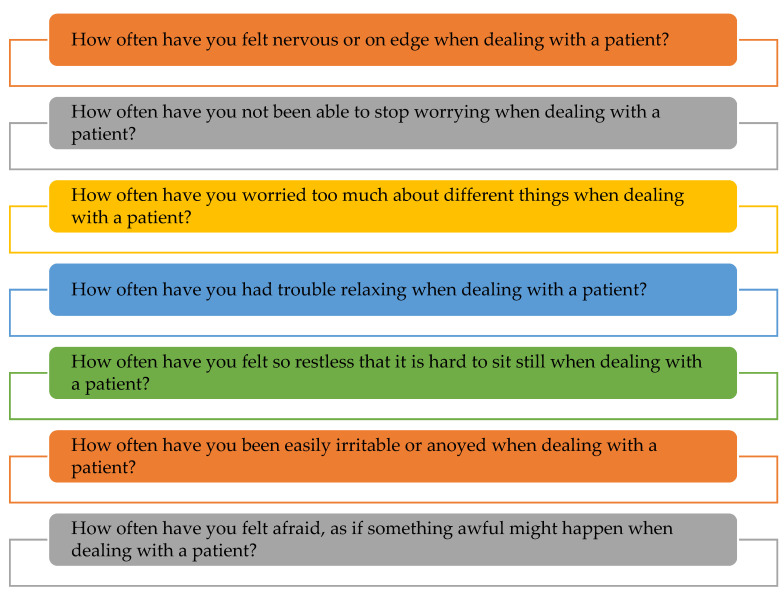
Modified GAD-7 scale adopted before and after vaccination in HCPs.

**Figure 2 vaccines-10-02076-f002:**
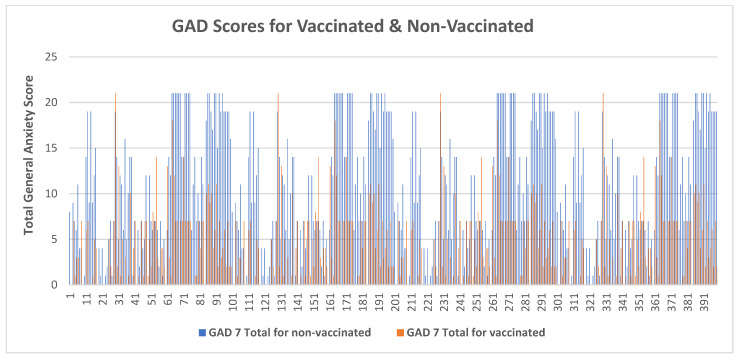
Comparison of anxiety scores of HCPs with and without vaccinations. **GAD:** Generalized Anxiety Disorder Assessment.

**Table 1 vaccines-10-02076-t001:** Distribution of demographic details of participants (*n* = 798).

Demographics	Vaccinated *n*%	Non-Vaccinated *n*%
Age		
10–20 years	10 (2.5)	9 (2.2)
21–30 years	295 (73.9)	303 (75.9)
31–40 years	27 (6.7)	30 (7.5)
41–50 years	42 (9.0)	34 (8.5)
Above 50 years	25 (6.2)	28 (7.0)
Mean age	24.32 ± 0.281	26.41 ± 0.493
Gender:		
Male	144 (36.09)	144 (36.0)
Female	255 (63.90)	255 (63.9)
Level of Education:		
Undergraduate	27 (6.0)	24 (6.0)
Graduate	316 (79.1)	323 (80.9)
Postgraduate	50 (12.5)	48 (12.0)
Below Undergraduate	6 (1.5)	4 (1.0)
Profession:		
Dentist	240 (60.1)	244 (61.15)
Medical Doctor	74 (11.77)	76 (19.0)
Nurse	85 (21.30)	79 (19.7)

**Table 2 vaccines-10-02076-t002:** Distribution of GAD-7 scores in unvaccinated individuals according to severity of anxiety (*n* = 399).

GAD-7 Scores for Non-Vaccinated	
	*n*%
Minimal	88 (21.9)
Mild	108 (26.9)
Moderate	80 (20.0)
Severe	123 (30.9)
Total	399 (100.0)

**Table 3 vaccines-10-02076-t003:** Distribution of GAD-7 scores in vaccinated individuals according to severity of anxiety (*n* = 399).

GAD-7 Scores for Vaccinated	
	*n*%
Minimal	227 (56.9)
Mild	124 (30.9)
Moderate	40 (10.0)
Severe	8 (2.0)
Total	399 (100.0)

**Table 4 vaccines-10-02076-t004:** Comparison of mean GAD-7 scores amongst the vaccinated and unvaccinated participants through paired *t*-test (*n* = 798).

Variables	Mean	Std. Deviation	*p*-Value
Unvaccinated	10.63	7.10	0.001
Vaccinated	4.38	4.37	0.001

**Table 5 vaccines-10-02076-t005:** Comparison of gender with GAD-7 scores with Mann–Whitney U test (*n* = 798).

Variables	Gender	N	Mean Rank	*p*-Value
Unvaccinated	Males	144	243.36	0.001
	Females	255	175.51	
Vaccinated	Males	144	223.58	0.001
	Females	255	186.68	

**Table 6 vaccines-10-02076-t006:** Relationship of gender, education, and profession with anxiety scores (*n* = 798).

Variables			Unvaccinated GAD-7 Score	Vaccination GAD-7 Score
Spearman’s rho	Gender	*p*-value Correlation Coefficient	0.001 ^b^ −0.284	0.002 ^b^ −0.156
	Education level	*p*-value Correlation Coefficient	0.005 ^b^ 0.013	0.037 ^f^ 0.010
	Profession	*p*-value Correlation Coefficient	0.001 ^b^ 0.028	0.093 0.084

^b^*p* < 0.00, ^f^
*p* ≤ 0.05.

## Data Availability

The data presented in this study are available on request from the corresponding author.

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
