# Peer review of "Comparison of General Anxiety among Healthcare Professionals before and after COVID-19 Vaccination"

_vaccines, 2022, doi:10.3390/vaccines10122076_

Round 1

Reviewer 1 Report

Remove minor spell mistakes

Author Response

Point to point author team response to reviewer comments

Thank you for reviewing our manuscript. The corrections recommended by the respected reviewers are addressed in different sections of the manuscript. The track change is on in the manuscript. The corrections are further highlighted with distinct color for clarity. The detailed response to the reviewer comment is described  below:

Reviewer 1

Comment 1: Remove minor spelling mistakes

Authors response: Thank you, we are humbled. The correction is done in different sections of the manuscript

Reviewer 2 Report

I consider the paper unsuitable for publication.  The writing and presentation of the paper are very deficient. The wording of the paper needs to be revised. There are grammatical errors and confusing expressions and sentences. There are repetitions in the text.

The term medical professionals leads to a misunderstanding.

Compliance with ethical requirements is not explained. It does not have approval by an ethics committee. It is not known whether the objectives of the study and how informed consent was obtained were explained to the respondents.

In my opinion, the study is poor and the methodology has numerous flaws.  In this sense, it would be necessary to have more information on the methodology followed in the selection of the healthcare professionals. It is not clear that the sample size is adequate as the total number of the population and its distribution according to the established groups are unknown.

Paramedic staff should not be considered as professionals. The distribution between the different groups should be explained.

The elaboration of the questionnaire must be sufficiently explained.

The inclusion and exclusion criteria in the study are not clear.

The differences between medical assistants, technicians, paramedical staff, etc. are not clear to the reader.

It should be explained how the biases that can be introduced when distributing the questionnaire using social media platforms have been avoided.

Descriptive data of the study participants (mean age, SD) must be presented.

The presentation of the results in the tables is poor.

Figure 2 is excessively confusing.

I consider the conclusions to be excessively generic and poor and are limited to a reiteration of the results that have been found.

Author Response

Point to point author team response to reviewer comments

Thank you for reviewing our manuscript. The corrections recommended by the respected reviewers are addressed in different sections of the manuscript. The track change is on in the manuscript. The corrections are further highlighted with distinct color for clarity. The detailed response to the reviewer comment is described  below:

Reviewer 2:

I consider the paper unsuitable for publication. The writing and presentation of the paper are very deficient.

Comment 1: The wording of the paper needs to be revised. There are grammatical errors and confusing expressions and sentences.

Author response: Thank you, corrected.

Comment 2: There are repetitions in the text.

Author response: Thank you, corrected.

Comment 3: The term medical professionals lead to a misunderstanding.

Author response: Thank you, corrected.

Comment 4: Compliance with ethical requirements is not explained. It does not have approval by an ethics committee.

Author response: Thank you, corrected. Page 3, Lines 104-106.

Comment 5: It is not known whether the objectives of the study and how informed consent was obtained were explained to the respondents

Author response: Thank you, corrected. Page 5, Lines 177-179.

Comment 6: In my opinion, the study is poor and the methodology has numerous flaws. In this sense, it would be necessary to have more information on the methodology followed in the selection of the healthcare professionals.

Author response: The methodology has been checked for improvement and further revisions. The corrections suggested in previous comments and any further needed has been done in the main document.

Comment 7: It is not clear that the sample size is adequate as the total number of the population and its distribution according to the established groups are unknown.

Author response:

  1. The sample size was calculated from a reference article and the calculated sample size does representing the population of healthcare professional in Karachi, Pakistan.
  2. However, considering the nature of the study, a survey needs a large sample size this shortcoming is already mentioned in the limitations of the study, discussion section.

Comment 8: Paramedic staff should not be considered as professionals.

Author response: Thank you, corrected.

Comment 9: The distribution between the different groups should be explained.

Author response: There are no groups in this study, there are different categories of healthcare professionals which are either vaccinated or non-vaccinated.

Comment 10: The elaboration of the questionnaire must be sufficiently explained.

Author response: Thank you, corrected. Page 3, Lines 122-137.

Comment 11: The inclusion and exclusion criteria in the study are not clear.

Author response: Thank you, corrected.

Comment 12: The differences between medical assistants, technicians, paramedical staff, etc. are not clear to the reader.

Author response: Thank you, corrected.

Comment 13: It should be explained how biases that can be introduced when distributing the questionnaire using social media platforms has been avoided.

Author response: Thank you, corrected. Page 3, Lines 139-140.

Comment 14: Descriptive data of the study participants (mean age, SD) must be presented.

Author response: The age data is presented in frequency and percentage, the study comprised of 5 age groups with a specific range. The mean and SD calculation is not possible and needed as such.

Comment 15: The presentation of the results in the tables is poor.

Author response: Thank you, corrected.

Comment 16: Figure 2 is excessively confusing.

Author response: Thank you, corrected. Page 7.

Comment 17: I consider the conclusions to be excessively generic and poor and are limited to a reiteration of the results that have been found.

Author response: Thank you, corrected. Page 11, Lines 411-415.

Reviewer 3 Report

In general, the content of this study is too simple, and the research design lacks certain depth and rigor. For example, the source of the included paticipants was not explained (for example, What size and type of institution is the research site; the majority of the included population is young people aged 21-30, which is not representative enough). In addition, the inclusion and exclusion criteria of the study were not strict. There are many wrong sentences and spaces that make the reader uncomfortable.

Author Response

Point to point author team response to reviewer comments

Thank you for reviewing our manuscript. The corrections recommended by the respected reviewers are addressed in different sections of the manuscript. The track change is on in the manuscript. The corrections are further highlighted with distinct color for clarity. The detailed response to the reviewer comment is described  below:

Reviewer 2:

In general, the content of this study is too simple, and the research design lacks certain depth and rigor.

Comment 1: For example, the source of the included participants was not explained (for example, what size and type of institution is the research site; the majority of the included population is young people aged 21-30, which is not representative enough).

Author response: Thank you, corrected. Page 3, Lines 116-121. The targeted hospital was in majority of young dental individuals who were easy to approach. In comparison to this age group, specialists and consultants who are normally of higher age were less in the hospital settings.

Comment 2: In addition, the inclusion and exclusion criteria of the study were not strict.

Author response: Thank you, corrected. Page 4, Lines 161-172.

Comment 3: There are many wrong sentences and spaces that

make the reader uncomfortable.

Author response: Thank you, corrected.

Round 2

Reviewer 2 Report

The paper has been improved, but there are still issues that need to be corrected.

There are typographical errors in the text. For example on page 3 line 138, on page 7 line 230, on page 8, line 271 and 272, on page 9, line 312...

In relation to the methodology of the study, some of the questions raised have been answered, but others remain unanswered. It would be necessary to have more information on the methodology followed in the selection of the healthcare professionals.

Descriptive data of the study participants (mean age, SD) must be presented.

The conclusions section remains excessively generic in relation to the objectives set out in the study.

Round 2

Point-to-point author team response to reviewer comments

Thank you for reviewing our manuscript. The corrections recommended by the respected reviewers are addressed in different sections of the manuscript. The track change is on in the manuscript. The corrections are further highlighted with distinct colors for clarity. The detailed response to the reviewer's comment is described below:

Reviewer 2:

The paper has been improved, but there are still issues that need to be corrected.

Comment 1: There are typographical errors in the text. For example, on page 3 line 138, on page 7 line 230, on page 8, lines 271 and 272, on page 9, line 312.

Author’s Response: Thank you, corrected.

Comment 2: In relation to the methodology of the study, some of the questions raised have been answered, but others remain unanswered. It would be necessary to have more information on the methodology followed in the selection of healthcare professionals. 

Author’s Response: Thank you corrected. Page 3, lines 109-118.

Comment 3: Descriptive data of the study participants (mean age, SD) must be presented.

Author’s Response: Thank you, corrected. Page 5, line 193,194.

Comment 4: The conclusions section remains excessively generic in relation to the objectives set out in the study.

Author’s Response: Thank you, corrected. Page 11, lines 401-411.

Reviewer 3 Report

There are several points that need to be explained or modified by the authors:

1. I can't understand the included objects. This study included 399 participants, while the number of vaccinated and unvaccinated people was 399 (table 2 and table 3), respectively. So this is a comparison between before and after vaccination for all participants? Or a comparison between 399 vaccinated participants and 399 unvaccinated participants? The logic of the whole study is very confused.

2. What the Figure 2 mean? Please describe it in the text.

3. Please modify unnecessary spaces or punctuation, such as line 137, 138 and 153.

4. Line 189, please change the "patients" into "participants".

Author Response

Point-to-point author team response to reviewer comments

Thank you for reviewing our manuscript. The corrections recommended by the respected reviewers are addressed in different sections of the manuscript. The track change is on in the manuscript. The corrections are further highlighted with distinct colors for clarity. The detailed response to the reviewer's comment is described below:

Reviewer 3:

There are several points that need to be explained or modified by the authors:

Comment 1: I can't understand the included objects. This study included 399 participants, while the number of vaccinated and unvaccinated people was 399 (table 2 and table 3), respectively. So, this is a comparison between before and after vaccination for all participants? Or a comparison between 399 vaccinated participants and 399 unvaccinated participants? The logic of the whole study is very confusing.

Author’s Response: Thank you, corrected. The total number of objects was 798, divided equally between vaccinated and unvaccinated. Page 5, line 197 and table 1.

Comment 2: What the Figure 2 mean? Please describe it in the text.

Author’s Response: thank you, corrected. Page 6 lines 213-217.

Comment 3: Please modify unnecessary spaces or punctuation, such as lines 137, 138, and 153.

Author’s Response: thank you, corrected

Comment 4: Line 189, please change the "patients" to "participants".

Author’s Response: Thank you, corrected. Page 5, line 241

Round 3

Reviewer 2 Report

The paper has been improved so that it can be published.

Author Response

Thanks for your suggestion.

Reviewer 3 Report

I have nothing to say

Author Response

Thanks for your suggestion.